# Optimal acceleration voltage for near-atomic resolution imaging of layer-stacked 2D polymer thin films

Baokun Liang[1,8], Yingying Zhang[2,8], Christopher Leist[1], Zhaowei Ou[3], Miroslav Položij[2], Zhiyong Wang[2], David Mücke[1], Renhao Dong[2,4], Zhikun Zheng[3], Thomas Heine[2,5,6], Xinliang Feng[2,7], Ute Kaiser[1✉] & Haoyuan Qi[1,2✉]

Despite superb instrumental resolution in modern transmission electron microscopes (TEM), high-resolution imaging of organic two-dimensional (2D) materials is a formidable task. Here, we present that the appropriate selection of the incident electron energy plays a crucial role in reducing the gap between achievable resolution in the image and the instrumental limit. Among a broad range of electron acceleration voltages (300 kV, 200 kV, 120 kV, and 80 kV) tested, we found that the highest resolution in the HRTEM image is achieved at 120 kV, which is 1.9 Å. In two imine-based 2D polymer thin films, unexpected molecular interstitial defects were unraveled. Their structural nature is identified with the aid of quantum mechanical calculations. Furthermore, the increased image resolution and enhanced image contrast at 120 kV enabled the detection of functional groups at the pore interfaces. The experimental setup has also been employed for an amorphous organic 2D material.

[1] Central Facility for Electron Microscopy, Electron Microscopy Group of Materials Science, Universität Ulm, 89081 Ulm, Germany. [2] Faculty of Chemistry and Food Chemistry & Center for Advancing Electronics Dresden (cfaed), Technische Universität Dresden, 01062 Dresden, Germany. [3] Key Laboratory for Polymeric Composite and Functional Materials of Ministry of Education, School of Chemistry, and State Key Laboratory of Optoelectronic Materials and Technologies, Sun Yat-sen University, 510275 Guangzhou, P. R. China. [4] Key Laboratory of Colloid and Interface Chemistry of the Ministry of Education, School of Chemistry and Chemical Engineering, Shandong University, 250100 Jinan, P. R. China. [5] Helmholtz Center Dresden-Rossendorf, Institute of Research Ecology, Leipzig Research Branch, 04318 Leipzig, Germany. [6] Department of Chemistry, Yonsei University, 03722 Seoul, Republic of Korea. [7] Max Planck Institute of Microstructure Physics, 06120 Halle (Saale), Germany. [8]These authors contributed equally: Baokun Liang, Yingying Zhang. ✉email: ute.kaiser@uni-ulm.de; haoyuan.qi@uni-ulm.de

From organic field-effect transistors (OFET) to organic solar cells, from gas filtration to catalysis, organic 2D crystals, such as 2D polymers and their layer-stacked structures of 2D covalent organic frameworks (COFs), are unfolding their potentials in a broad spectrum of novel applications[1,2]. Among the various research fields, structural elucidation is typically the key to a better understanding of structure-function correlations. Therefore, probing the internal construction of organic 2D crystals, ultimately down to the atomic scale, has been a long-sought goal of materials scientists. For instance, one of the intriguing characteristics of 2D polymers and 2D COFs is the designability of the pore interfaces[3]. Via direct polycondensation or pore surface engineering, selected side groups have been rationally incorporated onto the 2D polymer networks to enable functionalities, such as heterogeneous catalysis, proton/metal ion transport, energy storage, and gas adsorption[3]. Furthermore, the porosity can be fine-tuned by incorporating side groups with different physical sizes without altering the skeleton structure[4]. However, despite the advances in materials design, the precise characterization of pore interfaces remains a formidable task. Apart from highly crystalline organic 2D crystals, the structural understanding of amorphous organic 2D materials presents another substantial challenge due to the lack of proper characterization techniques.

Aberration-corrected high-resolution transmission electron microscopy (AC-HRTEM) is capable of direct imaging of atomic structures with sub-Ångström resolution[5–7]. However, electron irradiation damage often leads to instant disintegration of the molecular network during the imaging process[8]. The achievable image resolution on organic crystals is thus severely limited by specimen stability regardless of TEMs' optical performance. The lack of resolution precludes the visualization of the local structures such as lattice defects and grain boundaries[9–11] or delicate features like side groups/chains on the framework skeleton[3]. When it comes to amorphous organic 2D materials, due to the absence of long-range ordering, the magnitude of Bragg scattering is significantly reduced, leading to a decrease in signal-to-noise ratio, thus image visibility. For amorphous inorganic 2D materials, this issue can be circumvented by substantially increasing the electron fluence to obtain sufficient image contrast[12–15]. Nevertheless, this strategy becomes impractical for imaging organic amorphous 2D materials due to the severe electron irradiation damage.

Over the past decades, various experimental techniques have been designed to extract high-resolution information from 3D organic crystalline materials. The most commonly used technique, namely the low-dose approach, restricts the applied electron fluence below a critical value[16]. Structural information can then be extracted without causing substantial irradiation damage. When combined with direct electron detectors, high signal-to-noise ratios can be achieved even using an extremely low electron fluence[17,18]. Meanwhile, restraining the damage process presents another effective route toward better image resolution. For instance, sample vitrification[19,20] and/or encapsulation[21,22] could hinder the atom diffusion after bond scission, thereby facilitating bond reformation and enhancing the specimen lifetime under electron bombardment.

It is worth noting that, although high-resolution imaging of inorganic 2D materials could benefit substantially by restricting the electron energy below 80 keV[7], the rare TEM studies on organic 2D crystals are still carried out with 300 keV electrons[8]. Thus the exploration in the lower voltage regime remains scarce. The usage of high acceleration voltage is based on the fact that radiolysis (i.e., inelastic damage) is predominant in organic materials[23–25]. Since the inelastic scattering cross-section $\sigma_i$ is proportional to $1/\beta^2$ ($\beta = v/c$, $v$: electron velocity, $c$: speed of light), higher voltage is beneficial in reducing radiolysis. For conventional bulk samples with large thickness, the secondary electrons will induce a cascade of radiolysis events[26]. Thus, suppressing secondary electrons under high voltage was one of the main aims. Increasing the electron energy leads to a decrease of elastic scattering cross-section $\sigma_e$, which is also proportional to $1/\beta^2$. However, the ratio between elastic and inelastic scattering cross-section ($\sigma_e/\sigma_i$) decreases logarithmically with increasing incident electron energy[27]. Recently, Russo and co-workers have demonstrated that a 25% boost in $\sigma_e/\sigma_i$ can be achieved under 100 kV compared to conventional 300 kV[28]. Because elastically scattered electrons carry the structural information, increasing $\sigma_e/\sigma_i$ translates to a gain of structural information per unit damage[28]. Due to the low thickness of organic 2D crystals, the enhanced efficiency of electron usage may eventually outrun the detrimental effects of radiolysis damage (particularly from secondary electrons). This has inspired us to revisit the choice of acceleration voltage for imaging 2D polymer thin films.

In this work, we carry out systematic investigations to determine the optimal acceleration voltage for high-resolution imaging of two highly crystalline imine-based 2D polymer thin films (thickness up to 60 nm, Supplementary Fig. S1). The optimization considers the critical fluence and the proportion of the elastically scattered electrons under a broad range of electron energies (300 keV, 200 keV, 120 keV, 80 keV). Subsequently, AC-HRTEM is conducted at the optimal acceleration voltage with the aim to achieve improved image resolution and contrast on 2D polymers as well as amorphous organic 2D materials.

## Results and discussion

**Determination of the optimal electron acceleration voltage for 2D-PI-BPDA and 2D-PI-DhTPA.** Both 2D polyimine (PI) thin films were synthesized via the surfactant-monolayer-assistant interfacial synthesis (SMAIS) approach using 5,10,15,20-tetrakis(4-aminophenyl)-porphyrin (TAPP) as nodes with 4,4′-biphenyl-dicarboxaldehyde (BPDA) or 2,5-dihydroxyterephthalaldehyde (DhTPA) linkers[29,30]. Figure 1a presents the structural models of 2D-PI-BPDA and 2D-PI-DhTPA derived by density-functional tight-binding (DFTB) calculations. The selected area electron diffraction (SAED) patterns clearly demonstrate the high crystallinity of both specimens, and the measured lattice parameters of 30 Å (2D-PI-BPDA) and 25 Å (2D-PI-DhTPA) agree well with the theoretical values (Supplementary Fig. S2). In order to evaluate the efficacy of different electron energies, it is essential to determine the key factors, i.e., critical fluence and efficiency of electron usage during imaging.

The critical fluences of 2D-PI-BPDA and 2D-PI-DhTPA were measured by monitoring the fading of the reflections' intensity in the SAED pattern as the fluence accumulates[23,25]. At each applied acceleration voltage (80, 120, 200, 300 kV), the SAED series were taken under constant electron flux (0.48 e⁻/Å²s) at five different positions. As shown in Fig. 1b, after applying a fluence of 4.8 e⁻/Å² at 300 kV, 2D-PI-DhTPA maintained the high-resolution information up to the diffraction spot indexed 17 0 0, i.e., 1.4 Å. As the fluence accumulates, the degradation of long-range ordering under electron bombardment leads to the disappearance of the higher-order reflections and thus a deterioration of attainable image resolution. The critical fluence for a specific resolution can then be determined when the corresponding reflection intensity is reduced to a threshold ($1/e$) of its initial value.

Since the critical fluence analysis involves the evaluation of numerous SAED patterns, we have applied machine learning techniques for higher efficiency, accuracy, and reliability (see Methods and Supplementary Fig. S3). A U-Net type neural network was trained to automatically identify Bragg reflections in the SAED patterns and generate integrated intensity profiles (Fig. 1c). In order to enhance the data reliability, instead of

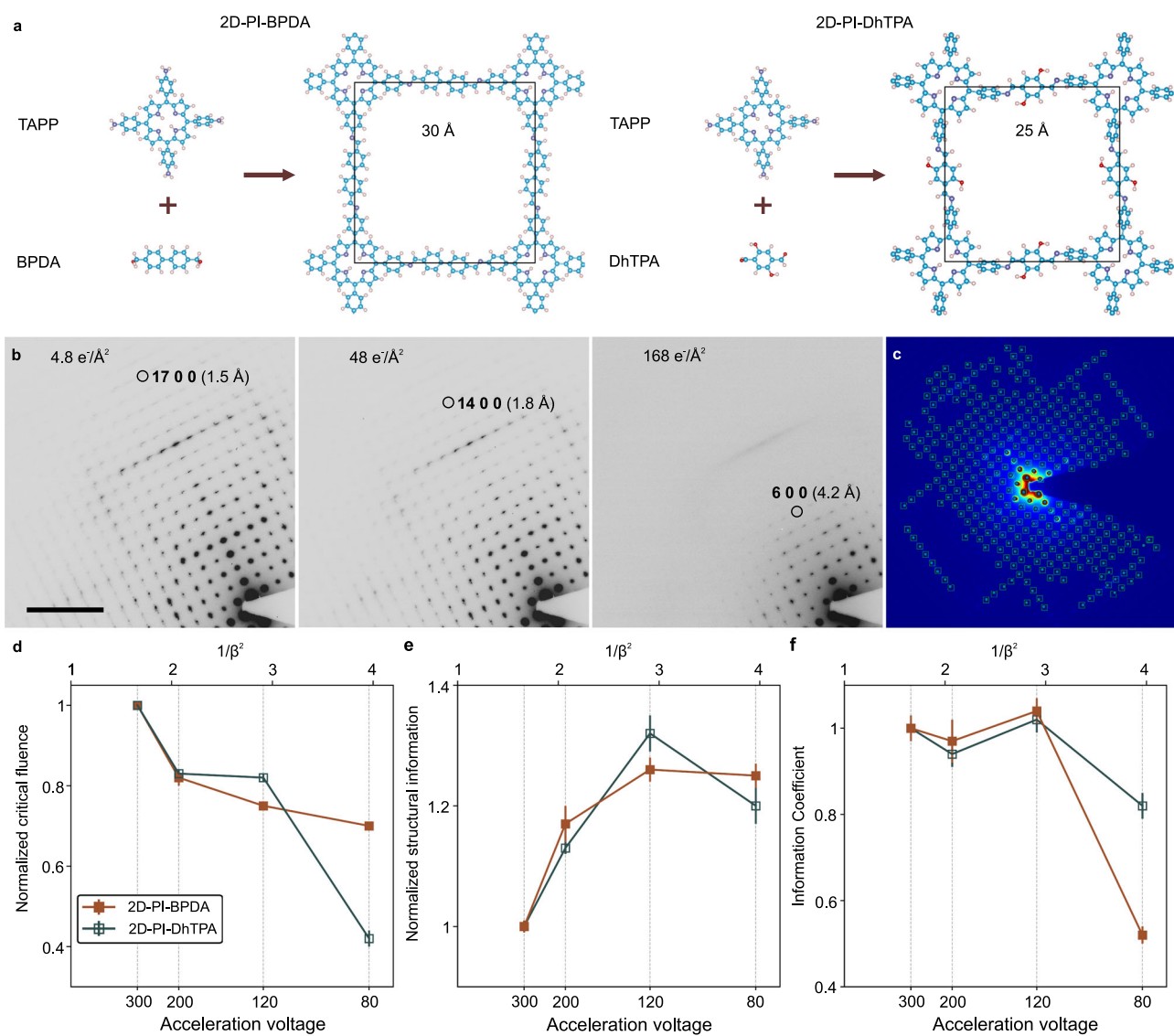

**Fig. 1 Schematic structure and acceleration voltage optimization of 2D-PI-BPDA and 2D-PI-DhTPA. a** The reaction scheme of 2D-PI-BPDA and 2D-PI-DhTPA (blue: C, purple: N, pink: H, red: O). **b** Selected images from a SAED pattern series on 2D-PI-DhTPA with a total fluence of 4.8, 48, and 168 e⁻/Å², respectively. Scale bar: 2 nm⁻¹. **c** Automatic identification of diffraction spots via the neural network. **d–f** Critical fluence for the resolution range from 5 to 6 nm⁻¹ (1.7–2.0 Å) (**d**) structural information proportion (**e**) and information coefficient (**f**) as a function of the acceleration voltage. All values are normalized to those at 300 kV. The error bar corresponds to the standard error of three measurements. The sienna and teal colored lines represent 2D-PI-BPDA and 2D-PI-DhTPA, respectively.

selecting single Friedel pairs, all reflection intensities inside a selected resolution range were integrated and plotted as a function of the electron fluence (Supplementary Fig. S4), offering a precise assessment of the critical fluence for desired resolution range. Figure 1d presents the critical fluence ($\delta_{cr}$) of 2D-PI-BPDA and 2D-PI-DhTPA as a function of the acceleration voltage. For clarity, the absolute fluence value under 300 kV has been normalized to unity. With decreasing acceleration voltage, the sample could withstand less electron fluence, which originates from the increased inelastic scattering and thus intensified radiolysis damage[24]. It would appear that a higher acceleration voltage would be beneficial, as reported in previous literature[23,24,31]. However, the critical fluence only specifies the total number of applicable electrons regardless of whether the electron carries structural information or not.

Since phase contrast in HRTEM imaging mode is determined by the interference between the forward scattered and the elastically scattered electrons (Bragg reflected beams), the proportion of the elastically scattered electrons within the critical fluence plays a vital role in the final image contrast. For quantification, we have specified the efficiency of electron usage ($\varepsilon_{el}$) as the ratio between the intensity of all Bragg reflections ($I_{Bragg}$) and the total integrated intensity ($I_{total}$) with the central beam included (i.e., $\varepsilon_{el} = I_{Bragg}/ I_{total}$). Since radiation damage will lead to the decrease of $I_{Bragg}$, giving rise to measurement error, the evaluation of $\varepsilon_{el}$ was carried out on SAED patterns acquired with a total fluence of merely 2 e⁻/Å². And the neural-network analysis was applied for data evaluation (see Methods and Supplementary Fig. S5). Figure 1e represents the normalized $\varepsilon_{el}$ as a function of the acceleration voltage. $\varepsilon_{el}$ increases with decreasing voltage and reaches a turning point at 120 kV. Under the kinematical theory, the elastic scattering cross-section $\sigma_e$ is proportional to $1/\beta^2$, suggesting a linear increase of $\varepsilon_{el}$ with decreasing voltage[8]. The linear relation has been experimentally demonstrated on graphene[28]. However, the kinematical scattering is valid only for very thin specimens, for which the Bragg

reflection intensity is negligibly small compared to that of the direct beam. Due to the finite thickness of the 2D polymer thin films (up to 60 nm), the kinematical theory no longer applies, which is illustrated by the non-linear change of $\varepsilon_{el}$ in Fig. 1e. Particularly at lower voltages where the elastic scattering mean free path is reduced, dynamical scattering leads to the intensity exchange between Bragg reflections and the direct beam[32]. In addition, increased inelastic events at lower voltage cause a further decrease of elastic scattering amplitude, giving rise to reduced $\varepsilon_{el}$ under 80 kV[32].

Due to the diverse behavior of $\delta_{cr}$ and $\varepsilon_{el}$, a trade-off between these two factors is necessary to pinpoint the optimal acceleration voltage for imaging 2D polymer thin films. For this purpose, we specified an 'information coefficient' to assess the total efficacy of electron energies with respect to obtaining the highest resolution of the image,

$$\zeta = \delta_{cr} \bullet \varepsilon_{el} \qquad (1)$$

Figure 1f shows the plots of $\zeta$ for 2D-PI-BPDA and 2D-PI-DhTPA for different acceleration voltages. It can be clearly seen that 120 kV offered the highest information coefficient in both materials. In other words, for a target resolution, the absolute number of information-carrying electrons is the highest under 120 kV, giving rise to enhanced signal-to-noise (S/N) ratio and image contrast. The boost in image contrast was further evidenced by image simulation (Fig. 2 and Supplementary Fig. S6). Supplementary Figure S7 shows the calculated thickness-defocus maps of three more 2D polymers, including 2D polyimide, 2D polyamide and viologen-immobilized 2D polymer[29,33], under 300 kV and 120 kV. As can be seen, compared with the conventional 300 kV, a contrast enhancement is expected with a 41–113 % increase when the sample thickness is below 10 nm.

Here, it is worth noting that the optimal acceleration voltage of 120 kV was experimentally determined on the 2D polymer thin films with thicknesses up to 60 nm, which is in good agreement

with the energy optimization for biological specimens with a given thickness[28]. Reducing the sample thickness would shift the optimal energy to even lower values[28] (Supplementary Fig. S6). On the other hand, lowering the acceleration voltage gives rise to more inelastic scattering events, which decreases the elastic scattering amplitude and thus phase contrast when a finite thickness is considered. If the inelastically scattered electrons undergo another elastic scattering process, the scattered and non-scattered parts are coherent and may provide crystal structure information[32]. However, due to chromatic aberration, the inelastically scattered electrons are no longer focused onto the Gaussian image plane, leading to image blurring[7,34]. Chromatic aberration correction elevates the resolution limit imposed by the temporal coherence damping function. Meanwhile, it enhances the image contrast due to the increased S/N ratio by refocusing the coherent inelastic electrons onto the Gaussian image plane[35]. The optimal electron energy for thinner specimens and the effects of chromatic aberration correction thus warrant further experimental investigations.

**AC-HRTEM imaging of 2D-PI-BPDA and 2D-PI-DhTPA under 120 kV.** Subsequently, we performed high-resolution imaging on both polymers under 120 kV. Figures 3a and 4a present the unprocessed AC-HRTEM images of 2D-PI-BPDA and 2D-PI-DhTPA, respectively. The images were acquired with an electron fluence for ca. 80 e⁻/Å², which is close to the pre-determined critical fluence of the resolution range 5–6 nm⁻¹ (1.7–2.0 Å). The FFT patterns (insets) reveal reflections up to the spatial frequency of 5 nm⁻¹, reaching up to 1.9 Å. Since a TEM image is a 2D projection of the 3D object, in order to reveal the structure of the 2D polymer film, the atomic models are derived by the DFTB method (Supplementary Fig. S8). As demonstrated in Figs. 3b and 4b, the simulated images based on the DFTB models well agree with the experimental ones, showing the TAPP

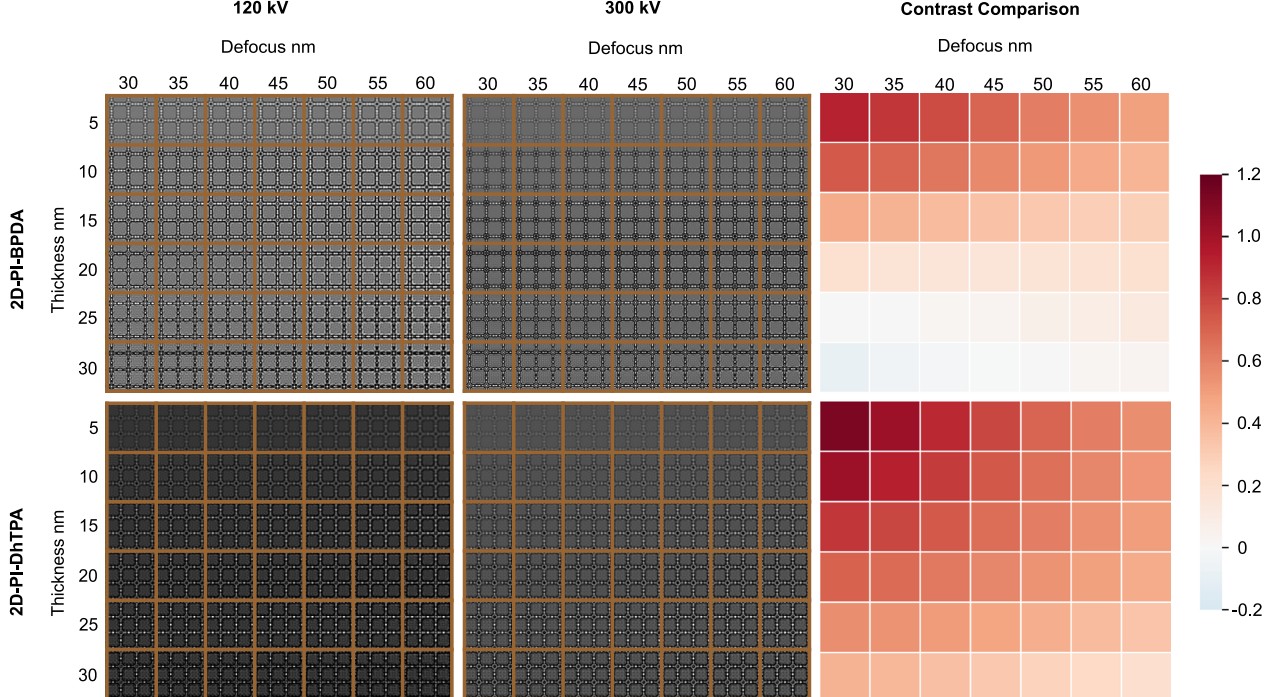

**Fig. 2 Contrast comparison between 300 kV and 120 kV via image simulation.** Simulated thickness-defocus maps at 120 and 300 kV of 2D-PI-BPDA and 2D-PI-DhTPA obtained by using multislice algorithm in QSTEM software. The coefficient of variation of the image grayscale is employed to represent the contrast. The contrast difference between 120 kV and 300 kV, i.e., (Contrast₁₂₀-Contrast₃₀₀)/Contrast₃₀₀, is presented by the heat maps in the third column.

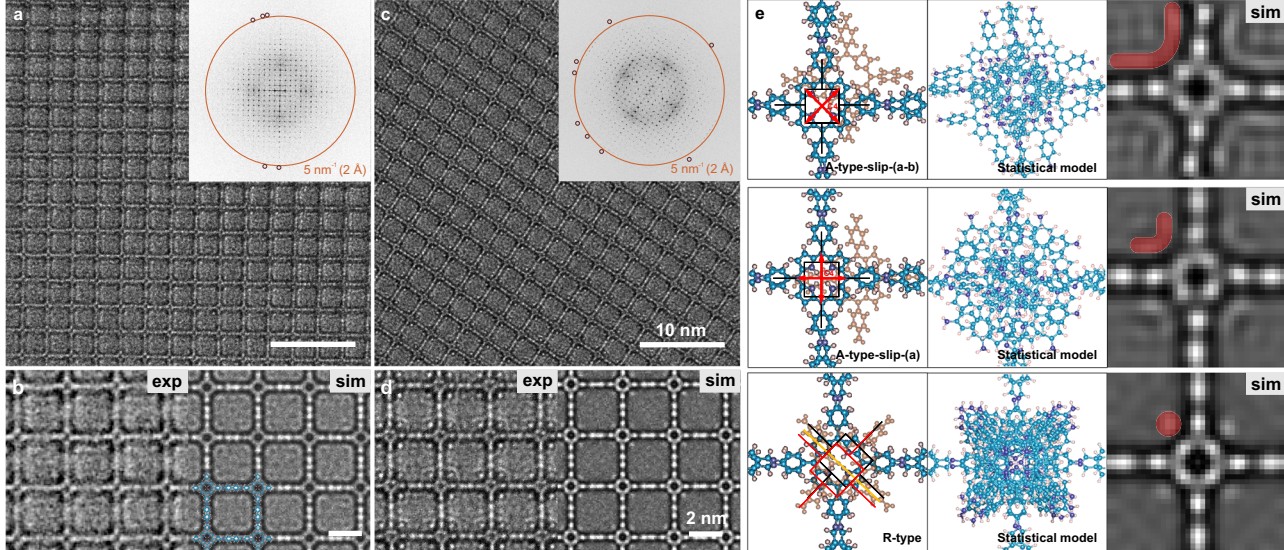

**Fig. 3 AC-HRTEM imaging of 2D-PI-BPDA and image simulations. a, c** Unprocessed AC-HRTEM images of 2D-PI-BPDA without and with TAPP interstitials, respectively. Scale bar: 10 nm. Insets: FFT patterns of (**a**) and (**c**). The orange circle marks the spatial frequency of 5 nm$^{-1}$ (i.e., 2 Å). The maroon circles mark the reflections beyond 5 nm$^{-1}$. **b, d** Left: magnified images from (**a**) and (**c**), respectively. The images have been denoised using Wiener filtering. Right: simulated images of 2D-PI-BPDA without molecular interstitials and with R-type TAPP interstitials. Simulation conditions: thickness 15 nm, defocus 50 nm. Scale bar: 2 nm. **e** Plausible stacking modes of TAPP interstitials derived by DFTB calculations. The statistical models and corresponding simulated images are shown in the middle and right column, respectively. The emerging contrast due to the interstitial TAPP molecules is marked in red on the simulated images.

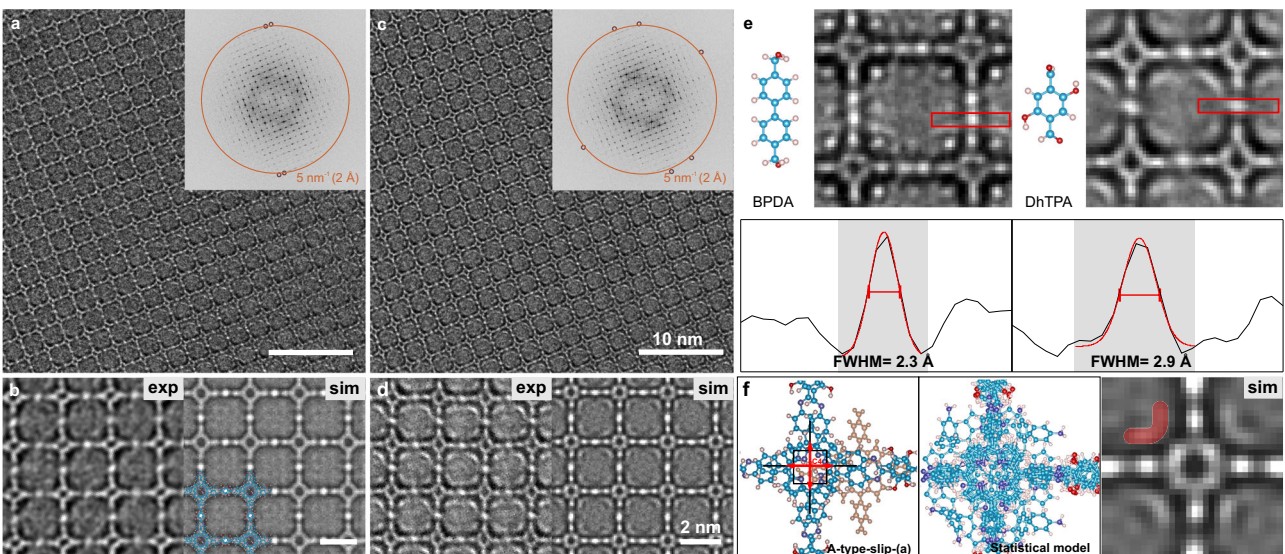

**Fig. 4 AC-HRTEM imaging, image simulation of 2D-PI-DhTPA, and comparison of the linker width with 2D-PI-BPDA. a, c** Unprocessed AC-HRTEM images of 2D-PI-DhTPA without and with TAPP interstitials, respectively. Scale bar: 10 nm. Insets: FFT patterns of (**a**) and (**c**). The orange circle marks the spatial frequency of 5 nm$^{-1}$ (i.e., 2 Å). The maroon circles mark the reflections beyond 5 nm$^{-1}$. **b, d** Left: magnified images from (**a**) and (**c**) respectively. The images have been denoised using Wiener filtering. Right: simulated images of 2D-PI-DhTPA without molecular interstitials and with A-type-slip-(*a*) TAPP interstitials. Simulation conditions: thickness 20 nm, defocus 40 nm. Scale bar: 2 nm. **e** Upper row: real-space-averaged images of 2D-PI-BPDA and 2D-PI-DhTPA, respectively. Lower row: integrated line profiles obtained from the boxed regions in the images. The FWHMs of the linker width are 2.3 Å and 2.9 Å, respectively. **f** DFTB models of A-type-slip-(*a*) TAPP interstitials in 2D-PI-DhTPA. The emerging contrast due to interstitials is marked in red on the simulated image.

nodes and BPDA/DhTPA linker molecules unambiguously. Note that the contrast enhancement under 120 kV enables image acquisition with low defocus values (i.e., 40–50 nm), thus significantly reducing contrast delocalization[36]. In other words, the image signal is sharply localized on the molecular framework without substantial blurring. Together with the sub-2-Å resolution, low delocalization allows the visualization of even

the 4-Å-sized pore on the porphyrin rings[37], which is not possible at 300 kV due to the necessity of higher defocus values for image visibility (Supplementary Fig. S9).

Interestingly, although the matching between experimental image and simulation is well maintained in most of the sample areas, we occasionally observe abnormal contrast in the vicinity of the porphyrin cores in both 2D polymer materials (Figs. 3c and 4c).

For instance, in 2D-PI-BPDA, we found four bright spots in the vicinity of the porphyrin node, which are positioned along the diagonal of the square lattice (Fig. 3d). However, despite our best efforts, the bright spots could not be reproduced in image simulation regardless of the applied parameters (thickness, defocus, aberrations). In other words, the abnormal contrast is not related to image processing and the microscope performance but rather associated with additional structural features in the specimen, which have remained elusive so far. The distance between two diagonal spots was measured to be 17.3 Å (Supplementary Fig. S10), which is close to the length between the diagonal amino groups on the TAPP molecule[38,39]. Since porphyrin and its derivatives may readily self-assemble into highly ordered aggregates[40,41], we conjectured that a low amount of TAPP molecules might have intercalated in the 2D polymer frameworks (i.e., molecular interstitial defects), giving rise to additional image contrast.

**Molecular interstitial defects and detection of side groups.** In order to explore the plausibility of molecular interstitials, we have conducted quantum mechanical calculations based on self-consistent charge (SCC)–DFTB theory. Additional TAPP molecules were inserted between the TAPP nodes in the 2D polymer framework. The DFTB calculations revealed two distinct types of the interstitial TAPPs; either aligned with the 2D polymer network and slid along the high symmetry axes, i.e., $a$ or $a$-$b$, denoted as A-type, or rotated with respect to the neighboring layers by 31° for 2D-PI-DhTPA and 42° for 2D-PI-BPDA and slightly slid, denoted as R-type (Fig. 3e, Supplementary Fig. S11). Note that an exactly eclipsed stacking is energetically unfavorable due to the electrostatic repulsion between TAPP molecules[42]. Furthermore, during the synthesis of 2D-PI-BPDA and 2D-PI-DhTPA (see Methods), the acidity constant $pK_a$ value of the TAPPs in the aqueous solution was below the protonation threshold[43], suggesting the presence of protons at the pyrroline-like rings[44,45] and thus enhanced repulsion between porphyrin cores. In the formation of face-to-face dimers of di-/tri-cationic porphyrin, the electrostatic repulsion could be minimized via a relative shift or a rotation between the molecules[42]. Due to the four-fold symmetry of the 2D polymer lattice, the possible shift directions of the interstitial TAPPs, e.g., $a$, $-a$, $b$, $-b$, are nearly isoenergetic (Supplementary Fig. S11). In order to create a realistic model of actual crystal for HRTEM simulation, we built a model with a statistical distribution of all equivalent possibilities. A 120-layer model was constructed with 1/3 of the layers replaced by interstitial TAPPs. Each TAPP was randomly assigned with one of the equivalent shifts (Supplementary Fig. S12).

Figure 3e illustrates the statistical models, presenting possible molecular configurations of interstitial TAPPs in 2D-PI-BPDA. As demonstrated by the image simulation, the interstitials give rise to extra contrast in the vicinity of the porphyrin cores (Fig. 3e). To our delight, the bright spots could be well reproduced with the R-type configuration, suggesting the presence of rotated TAPPs intercalated within the 2D-PI-BPDA framework. It is worth noting that we did not observe any A-type interstitials in the experimental images, which might be attributed to their higher relative energy as compared to R-type (Supplementary Table S1). In contrast to 2D-PI-BPDA, A-type interstitial defects were discovered in 2D-PI-DhTPA, leading to crescent contrast near the porphyrin nodes (Fig. 4c, f). The energetic preference of A-type in 2D-PI-DhTPA was further confirmed by theoretical calculations (Supplementary Table S1).

Owing to the enhanced resolution and contrast under 120 kV, we were able to not only identify the discrepancy between the observed and expected 2D polymer structures but also uncover the presence of molecular interstitial defects, which have not been

reported so far. Due to the amino groups on TAPPs, the interstitials may alter the local chemical environment. Furthermore, the presence of interstitials could also lead to an expansion of the distance between the 2D polymer layers (Supplementary Fig. S11), thus reducing the density of atomic sites (i.e., OH, C=N, C–H) at the pore interface. The next question would be: can we extend 120 kV imaging to the observation of functional groups on the molecular skeleton?

Comparing the structure of 2D-PI-DhTPA and 2D-PI-BPDA, the main distinction lies in the linker molecules. DhTPA contains two hydroxyl groups, whereas the BPDA does not have any side chains. By plotting the integrated line profiles across the center of the linker molecules, we have measured the FWHM of the linker intensities in both 2D polymers (Fig. 4e). Interestingly, 2D-PI-DhTPA and 2D-PI-BPDA showed a clear difference in the linker widths (Fig. 4e). Statistical analysis averaging 100 linker sites revealed that the FWHM of DhTPA is ca. 70 % (0.9 Å) broader than that of BPDA, which can be associated with the presence of hydroxyl groups. We believe that these results open up an exciting opportunity for direct imaging and distinction of functional groups on the molecular skeleton, particularly when longer and more complicated side chains are present, such as carboxylic acids and phenyl groups. The direct observation of functional groups at the pore interface is highly desirable to probe the structure-property correlation, e.g., host-guest interactions. However, multifold challenges remain. For example, although pore surface engineering has been well established in 2D COFs in powder form[3], exfoliation of organic crystals down to a few or a few tens of nanometres while maintaining the pristine long-range order is not a trivial task[46]. Furthermore, the functionalization (via direct polymerization or post-synthetic modification) of interfacial-synthesized 2D polymer thin films is still in its infancy and remains to be explored.

**Visualization and quantification of short-range-order in amorphous organic 2D materials.** Since 120 kV could provide a higher information coefficient, we envisage that our experimental setup would also benefit the visualization of short-range-order in organic 2D materials due to the enhanced image contrast. To demonstrate this, we applied 120 kV imaging on an amorphous polyimine thin film ($a$-PI, Fig. 5c), which is an amorphous analog of the viologen-immobilized 2D polymer, synthesized via Schiff-base polycondensation on the air-water interface[33]. The SAED pattern revealed diffusive 100 and 200 rings (Fig. 5b, Supplementary Fig. S2), indicating the lack of long-range order and thus low scattering power. The low-magnification TEM image (Fig. 5e) illustrates that the $a$-PI film consists of a highly disordered network, evidencing its amorphous nature. In order to gain a better insight into the local structure of $a$-V2DP, AC-HRTEM imaging with sub-unit-cell resolution has been performed (Fig. 5c). The molecular network could be resolved with the bright spots corresponding to the node positions (Supplementary Fig. S13). We found that $a$-PI was composed of a mixture of expected hexagons and defective pentagons and heptagons. To achieve a quantitative analysis of the short-range order in $a$-PI, we have developed a neural network based on U-net architecture to determine node positions automatically (see Methods). Based on that, the length and angle between the nodes, nearest neighbor distribution (Supplementary Fig. S14), as well as the real-space mapping (Fig. 5d), and percentage of polygons (Fig. 5f) can be extracted statistically. The combination of high image contrast and machine learning has allowed us to establish a quantitative structural profile of the $a$-PI (Supplementary Table S2), which may pave the way for understanding structural signatures between different organic $a$−2D materials in the future.

In this work, we systematically evaluated the electron accelerating voltages for AC-HRTEM imaging of 2D polymer thin films,

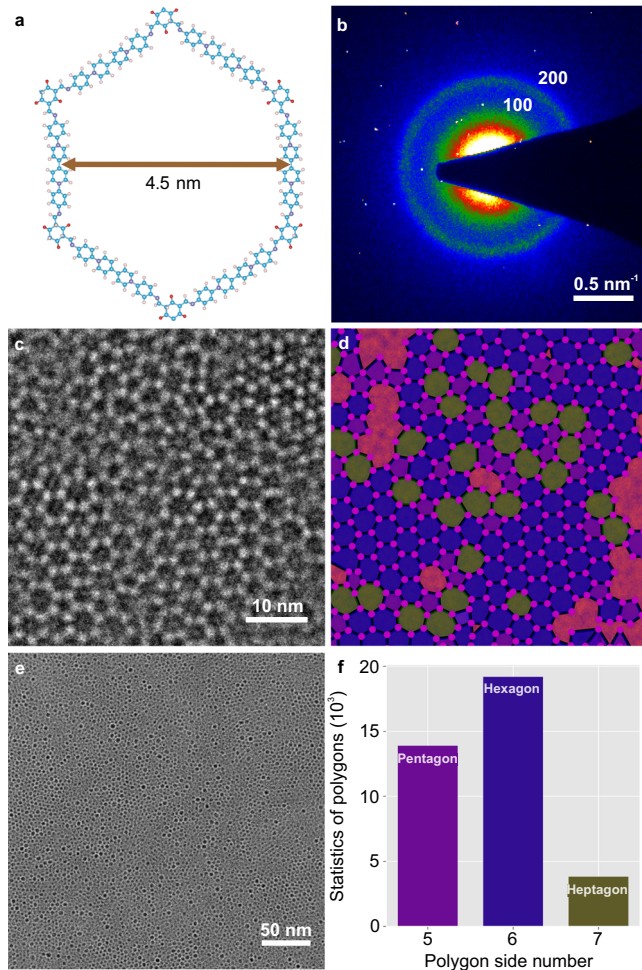

**Fig. 5 TEM characterization and statistical structural analysis of *a*-PI.**
**a** Atomic structure of the *a*-PI, the pore diameter is 4.5 nm. **b** SAED pattern of *a*-PI showing faint diffraction rings only up to the second order. **c** AC-HRTEM image of *a*-PI showing the lack of long-range order. The bright spots correspond to the node molecules. Scale bar: 10 nm. **d** Neural network polygon mapping of (**c**). The pink spots mark the node molecule positions, and the polygons are colored according to their number of sides (pentagon: purple; hexagon: blue; heptagon: green). **e** Overview TEM image demonstrating the presence of different polygon types on a larger scale. Scale bar: 50 nm. **f** Statistical results of the polygon distribution from a data set of nine AC-HRTEM images, each having a field-of-view of 200 nm².

exemplified by 2D-PI-BPDA and 2D-PI-DhTPA. We reached an image resolution of 1.9 Å at 120 kV. This improvement in image resolution has been achieved by maximizing the efficiency of electron usage, among the acceleration voltages of 300 kV, 200 kV, 120 kV, and 80 kV. As not only the resolution but also the image contrast is enhanced at 120 kV, sufficient structure visibility could be obtained under low defocus values (i.e., 40–50 nm) to mitigate the detrimental effects of contrast delocalization/blurring. This allows one-to-one mapping of the framework structures. Not only could the porphyrin pores be clearly resolved, but linkers with and without hydroxyl groups could also be distinguished. Surprisingly, we occasionally observed abnormal contrast in the vicinity of the porphyrin cores in both materials, which conflicts with their previously reported structures. Quantum mechanical calculations revealed that the additional contrast could be attributed to molecular interstitial defects – a defect type that had not been discovered in 2D polymers before. We envisage that our results will bring new insights into the defect types and pore interfaces in

organic 2D crystals and promote a deeper understanding of pore engineering in future studies. Besides imaging highly crystalline 2D polymers, employing the optimized acceleration voltage also allowed the elucidation of the structures in amorphous organic 2D materials. Image analysis via a U-net-based neural network provided access to a full spectrum of datasets, enabling quantitative description of medium and short-range-order in 2D crystalline and amorphous organic thin films.

## Methods

**Transmission electron microscopy.** TEM experiments were performed on an image-side aberration-corrected FEI Titan 80-300 operated at 80, 120, 300 kV, and a Thermo Fisher Talos operated at 200 kV. The FEI Titan is equipped with a CEOS hexapole aberration-corrector that corrects the geometrical axial aberrations up to the 3rd-order. Data acquisition was conducted on a Gatan UltraScan CCD camera. On FEI Talos the data acquisition was conducted on a Ceta CMOS camera.

*Electron diffraction.* Electron diffraction was performed both in selected-area electron diffraction (SAED) mode. For SAED acquisition, we used a selected-area aperture with a physical diameter of 50 μm, corresponding to a diameter of 480 nm in the image plane.

*AC-HRTEM imaging.* AC-HRETM imaging is divided into three steps, i.e., search, focus, and acquisition. Throughout the experiment, the pre-specimen shutter was applied, which automatically blanks the electron beam when camera viewing is deactivated. To search for a point of interest while preventing significant radiation damage, we selected an electron flux of 0.2 e⁻/Å²s. The large pixel size of 41.62 Å was used to maximize the number of electrons per pixel. We further enhanced the contrast by bright-field TEM imaging with the objective aperture (physical diameter: 40 μm, cut-off frequency: 3.6 nm⁻¹). After finding a point of interest, we removed the objective aperture and configured the imaging conditions (i.e., magnification, beam intensity, etc.) for focus and acquisition while keeping the beam blanked. We chose a pixel size of 0.836 Å, allowing information transfer to 1.67 Å. The electron flux was 420 e⁻/Å²s, and the beam diameter was 230 nm. For focus, the specimen stage was laterally shifted by 1–2 μm to avoid pre-exposure to the point of interest. After tuning the defocus to ca. 30 nm (judged by the Thon rings in FFT after amorphization of the 2D polymer film within the field-of-view), we moved back to the point-of-interest and acquired a single image with a total fluence of 76 e⁻/Å² (flux: 420 e⁻/Å²s, acquisition time: 0.18 s). HRTEM image simulation was carried out using QSTEM software[47].

*Critical fluence analysis.* SAED series were taken under constant electron flux at each applied acceleration voltage (80, 120, 200, 300 kV). As the fluence accumulates, the reflections gradually vanish from high to low orders. For data analysis, the high-order reflections at 2–6 nm⁻¹ are divided into four resolution ranges. To compensate for possible slight mis-tilt off the main zone axis of the illuminated area, the reflections in each resolution range are integrated, and each experiment is repeated in three areas. The critical fluence analysis involves the processing of numerous SAED patterns. For efficient and practical analysis of the image series, machine learning is applied. A U-Net type neural network is trained to recognize and pre-select the reflections on SAED images. The identified reflection spots' intensities are integrated inside the respective resolution range in a MATLAB script. The MATLAB script is from Tatiana Gorelik, the neural network implementation by Christopher Leist. The fading behavior of the reflection in a defined resolution range is recorded and plotted against the electron fluence, exemplified in Fig. S4. The critical fluence is defined when the intensity is reduced to a threshold (conventionally 1/*e*) of the initial value[19]. The respective damage cross-section is defined as the reciprocal of the critical fluence[48].

**Density functional tight-biding calculations.** Structural simulations were conducted using the self-consistent charge density-functional tight-binding (SCC-DFTB)[49] as implemented in Amsterdam Density Functional (ADF)-DFTB 2019 program suite (www.scm.com). The 3ob-3-1 parameter set for the X-Y element pair interaction (X, Y = C, H, N, O) was employed. All structures were optimized with D3 (BJ) dispersion correction. The multi-layered crystal models with a statistical distribution of interstitial defects were built using an in-house python code.

**Machine learning.** In order to reduce manual work in SAED image series evaluation, machine learning technique was applied to pre-select reflection spots in the image. A neural network of U-net type[50] was trained and integrated into the Matlab script to speed up the analysis process.

For the network training, five down-sampled 512 × 512 pixel SAED images with readily marked reflection spots were used. The labels form a binary map of the same size as the image. In our case, relatively low numbers of training images were used. Therefore instead of a traditional three down- and up-sampling steps, a 2-step network was applied. The smaller network has the benefit of regularization and prevents overfitting. The structure of the network is shown in Supplementary Fig. S3.

To enable the statistical analysis of the polygons in AC-HRTEM images, three networks were trained on simulated images showing semi-random spot positions on the same scale as the real AC-HRTEM images. The spot positions were determined by the method described in[51] for simulating semi-random atomic layers. These neural networks are of U-net type with three down- and up-sampling steps. They were trained with training sets consisting of 1000 to 4000 images. Three separate networks were used to find the spot positions, the positions of polygon centers, and segment regions unusable for evaluation.

A binary map was applied to the output of the neural network, from which the spot and polygon center positions were extracted. A Delaunay triangulation between the spot and center positions was applied. The bonds were isolated by subsequent removal of the connections to the center points, allowing the determination of bond lengths and angles. The statistical analysis was plotted, and a Gaussian fitting has been applied to determine the full width at half maximum (FWHM). Meanwhile, the center positions and their connections to nearby node molecules specified the type of polymer rings for the polygon mapping.

**Synthesis of 2D-PI samples**. *2D-PI-BPDA* was synthesized through SMAIS method in a petri dish with a diameter of 60 mm and a height of 17 mm. 25 ml Milli-Q water was injected. An aqueous solution of polystyrene sulfonate (PSS, 60 µL, 1 mg/mL) was then spread at the air-water interface and left undisturbed for 30 min. for the maximum diffusion. After the diffusion of PSS, 1 M p-toluenesulfonic acid monohydrate (PTSA) solution of 5,10,15,20-tetrakis(4-aminophenyl)-21H,23H-porphyrin (TAPP, 200 µL, 0.3 µmol) was injected into to the bottom of the solution. After 1 h, the aqueous solution of aldehyde 4,4'-Biphenyl-dicarboxaldehyde (BPDA, 4.72 ml, 1.8 µmol) was slowly injected into the bottom of the solution. The interfacial reaction was kept at room temperature for seven days. The products were transferred to Quantifoil TEM grids. Further details of the synthesis process can be found in Ou, Z. et al.[30].

*2D-PI-DhTPA* was synthesized through SMAIS method in a petri dish with a diameter of 60 mm and a height of 17 mm. 26 mL Milli-Q water was added. Then an aqueous solution of PSS (60 µL, 1 mg/mL) was spread at the air-water interface and left undisturbed for 30 min. Subsequently, 2 M PTSA solution of TAPP (200 µL, 0.3 µmol) was injected into the water phase. The monomer was allowed for dispersion in water for one hour, and then 2,5-dihydroxyterephthalaldehyde solution (DhTPA, monomer 2, 3.69 ml, 1.8 µmol) was added to the water phase. The interfacial reaction was kept static at room temperature for seven days. The products were transferred to Quantifoil TEM grids. Further details of the synthesis process can be found in Ou, Z. et al.[30].

*V2DP* was synthesized through SMAIS method. First, 50 mL Milli-Q water was injected into a beaker (160 mL, diameter = 12 cm) to form a static air/water interface. Then, 20 µL SOS (1 mg/mL in chloroform) was spread onto the water surface. After 10 min, 2 mL trifluoromethanesulfonic acid (TfOH) (7.4 µmol) aqueous solution of 1,1'-bis(4-aminophenyl)-[4,4'-bipyridine]-1,1'-diium chloride (4.8 µmol) was gently added using the syringe. After 30 min, 2 mL aqueous solution of 2,4,6-trihydroxybenzene-1,3,5-tricarbaldehyde (3.2 µmol) was added. The reaction was then kept undisturbed at room temperature for six days. The synthetic yellow film was deposited onto the TEM Quantifoil grid by the horizontal dipping method. The substrate with the a-V2DP film was dried at 80 °C for 30 min and rinsed with ethanol, Milli-Q water, and then dried in $N_2$ flow. Further details of the synthesis process can be found in Wang, Z. et al.[33].

## Data availability
The data supporting the findings of this study are available within the paper and its supplementary information files. Source data are provided with this paper. The DFTB models and the crystallographic information files for the structures reported in this paper have been deposited at the Zenodo under deposition no. 6481003. These data can be obtained free of charge via https://doi.org/10.5281/zenodo.6481003 Source data are provided with this paper.

## Code availability
The Python code for generating statistical models of molecular interstitials is available at https://doi.org/10.5281/zenodo.6481003. The Matlab script with neural network code for SAED pattern analysis and critical fluence determination is available at https://doi.org/10.5281/zenodo.6470239.

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

## Acknowledgements

B.L., C.L., D.M., U.K., and H.Q. gratefully acknowledge the funding from Deutsche Forschungsgemeinschaft (DFG, German Research Foundation) – 492191310; 426572620; 417590517 (SFB-1415), as well as the financial support the European Union's Horizon2020 research and innovation program under Grant Agreement No. 881603 (GrapheneCore3). Y.Z., M.P., and T.H. thank the ZIH Dresden for computer time. Y.Z. acknowledges China Scholarship Council. M.P. also thanks the financial support from the Alexander von Humboldt Foundation. Z.O., and Z.Z. thanks the financial support from the National Natural Science Foundation of China (51873236 and 51833011). Z.W., R.D., and X.F. acknowledge the financial support from the ERC Consolidator Grant on T2DCP (no. 819698), the ERC Starting Grant on FC2DMOF (no. 852909), the EU Graphene Flagship (no. 881603), the COORNET (SPP 1928), the DFG project (2D polyanilines, no. 426572620), the DFG SFB 1415 (no. 417590517), the CONJUGATION-706082, as well as the German Science Council, Center for Advancing Electronics Dresden (cfaed), EXC1056, and OR 349/1.

## Author contributions

B.L. performed the TEM experiments under the guidance of U.K. and H.Q. H.Q. conceived and designed the TEM experiments. B.L., C.L., D.M., U.K. and H.Q. analyzed and/or discussed the TEM data. Y.Z., M.P., and T.H. performed the DFTB calculations and analyzed the data. C.L. developed and applied the neural-networks for TEM data analysis. Z.O., Z.W., R.D., Z.Z., and X.F. synthesized the 2D polymers. B.L., U.K., and H.Q. wrote the manuscript with contributions from Y.Z., C.L., M.P. and extended comments from T.H. and X.F. All authors commented on the manuscript.

## Funding

## Competing interests

The authors declare no competing interests.
