## [Peer Review File · Nature Communications]

Optimal acceleration voltage for near-atomic resolution imaging of layer-stacked 2D polymer thin filmsREVIEWER COMMENTS

Reviewer #1 (Remarks to the Author):

In this manuscript, TEM has been pushed to its limits in order to visualize with high spatial resolution, crystalline as well as non-crystalline porous polymers. Surprisingly, it was possible to gain information on the difference in chemical nature (presence of functional groups) of the linker units. This is an important step in the development of a characterization tool box for the structural and chemical evaluation of the important class of porous covalent 2D polymers.

I recommend to

- 1) add the sample preparation protocol to the Methods and the supporting information.
- 2) add a discussion on the limitations and prospects of the approach to distinguish between different functional groups (and the potential radiation damage which may depend on the functional group).

Reviewer #2 (Remarks to the Author):

The authors describe an investigation of the electron acceleration voltage vs image resolution applied to imaging / structural determination of 2D Covalent Organic Frameworks (COFs). The results are highly significant and this study will assist future researchers in interrogating 2D organic materials via TEM.

The study is thoroughly undertaken and well-presented. The conclusions are supported by the experiments undertaken.

The most obvious deficiency (and it is a strong and unFAIR deficiency) is in the final two lines of the paper, in the Data Availability Statement. The zip file provided for review contained only the docx of the article and pdf of the supplementary information. At the very least, the computed structural models should be provided (i.e. the data upon which Fig S7 and S10 rely on) just like they were crystal structures. A representative model for the amorphous α -PI should also be included. The code and ML model could also be uploaded to zenodo or github and referenced appropriately.

At a couple of points the internal referencing has gone awry:

Page 6 Line 249, 250: "2D-PI" could refer to either of the two square lattice COFs that form the bulk of the study, 2D-PI-BPDA and 2D-PI-DhTPA (I assume it's the second of these). Also, the reference to Fig 3C and 3F must be incorrect, I assume they mean Fig 4C and 4F.

Similarly;

Page 6 Line 277,279,288: "Fig 4B/E/F" should be Figure 5X.

Fig S9 caption: "first-principle calculations" is a bit vague. If they used the same DFTB method as in the rest of the paper, they should say so here.

Reviewer #3 (Remarks to the Author):

This is an interesting study examining the optimization of the accelerating voltage for phase contrast HRTEM of beam sensitive 2D polymer thin films, inspired by the work of Russo and co-workers who also examined the idea of enhancing the information per electron by lowering the accelerating voltage. The authors show a significant advantage to working at 120 kV for these materials.

A few comments I feel would be good to address before publication:

What was not so clear to me is why exactly 120 kV turned out to be much better than 80 kV in this

case. This seems likely a combination of the sample thickness and chromatic aberrations. The authors indeed seem to speculate that the drop in the measured Eel at 80 kV is due to a combination of decreased scattering mean free paths and dynamic scattering, but it would be beneficial if they elucidate the reason for the drop below 120 kV more clearly. As it is, I see no mention of chromatic aberrations at all, and one would expect this to be quite important at lower kVs. How much of a role do chromatic aberrations play?

The authors describe the samples as being of tens of nanometers thick, up to 60 nm. That seems quite a large range of thicknesses. The findings of Russo et al (ref 28 of the present manuscript, eg their fig 5) would seem to suggest that quite a range of voltages might be optimal for the different tens of nanometers of thickness of the "2D" sheets considered here. Do the authors disagree with this? Is 120 kV optimal for all the different thicknesses of these 2D sheets?

The comparison shown between 300 kV and 120 kV in figure 2 is interesting. Would it not be useful to include also the other voltages considered, 200 and 80 kV to help more clearly say where the advantages exist (or not) for 120 kV compared to these as well?

A point-by-point response to reviewers' comments

Reviewer #1 (Remarks to the Author):

In this manuscript, TEM has been pushed to its limits in order to visualize with high spatial resolution, crystalline as well as non-crystalline porous polymers. Surprisingly, it was possible to gain information on the difference in chemical nature (presence of functional groups) of the linker units. This is an important step in the development of a characterization tool box for the structural and chemical evaluation of the important class of porous covalent 2D polymers.

Response: We are grateful for the reviewer's positive comments and constructive suggestions.

I recommend to

1) add the sample preparation protocol to the Methods and the supporting information.

Response: The reviewer's comment is highly appreciated. We have included the synthetic details in the Methods section of the revised manuscript (Page 9, subsection: Synthesis of 2D-PI samples).

2) add a discussion on the limitations and prospects of the approach to distinguish between different functional groups (and the potential radiation damage which may depend on the functional group).

Response: We thank the reviewer for the constructive comments. The engineering of the pore surface of the 2D polymer samples has attracted high research interest. By anchoring functional groups on the linker monomers, the pore interface properties are modified to achieve higher selectivity and storage capacity of target gas molecules^{1,2}. The direct observation of functional groups at the pore interface is highly desirable to probe the structure-property correlation, e.g., host-guest interactions. The current challenge mainly stems from the strict requirements in sample morphology and crystallinity for near-atomic-scale HRTEM imaging.

In order to access the high-resolution structural details, 2D polymer thin films with high crystallinity are required. Although the SMAIS method has been proven a robust route for producing thin 2D polymer films with high crystallinity, the functionalization (either via direct polymerization or pore surface engineering) is still in its infancy and remains to be explored. On the other hand, pore design and interface engineering have been well established in powder-form COF specimens². Yet, exfoliation of organic crystals down to a few tens of nanometers (required for HRTEM imaging) is not a trivial task. Particularly, the exfoliation process may already lead to the decay of long-range order. Since HRTEM images are interference patterns, the reduced long-range order poses additional limitations on the achievable image resolution. In the revised manuscript, we have added a discussion on the current challenges and future perspectives on direct imaging of functional groups (Page 6, lines 270 - 277).

The introduction of different chemical groups in the 2D polymer framework is expected to influence the sample stability under electron beam irradiation. In the last decades, significant efforts have been devoted to elucidating the correlation between critical fluence and sample structure. For example, aromatic compounds are more electron resilient than aliphatics due to π electron delocalization and conjugation³⁻⁵, suggesting higher stability of aromatic functional groups. In addition, organic polymers are prone to crosslinking after carbon-hydrogen bond scission, which leads to the loss of long-range order^{6,7}. Exchanging protium for chlorine on coronene molecules has increased the crosslinking dose by two orders of magnitude due to the reduced displacement cross-section of chlorine⁸, indicating a potential negative correlation between hydrogen content at the pore interface and the framework and side-chain stability. Quantification of functional group effects in electron resilience of 2D polymers is our next target. In the follow-up study, we will explore the functionalization of interfacial-synthesized 2D polymer thin films to systematically probe the influence of side groups on the framework stability.

Reviewer #2 (Remarks to the Author):

The authors describe an investigation of the electron acceleration voltage vs image resolution applied to imaging / structural determination of 2D Covalent Organic Frameworks (COFs). The results are highly significant and this study will assist future researchers in interrogating 2D organic materials via TEM. The study is thoroughly undertaken and well-presented. The conclusions are supported by the experiments undertaken.

Response: The reviewer's positive feedback is highly appreciated.

The most obvious deficiency (and it is a strong and unFAIR deficiency) is in the final two lines of the paper, in the Data Availability Statement. The zip file provided for review contained only the docx of the article and pdf of the supplementary information. At the very least, the computed structural models should be provided (i.e. the data upon which Fig S7 and S10 rely on) just like they were crystal structures. A representative model for the amorphous α -PI should also be included. The code and ML model could also be uploaded to zenodo or github and referenced appropriately.

Response: We thank the reviewer for pointing it out. As recommended by the referee, the computed CIF files have been uploaded to the Zenodo repository, which includes the DFTB models of:

- 2D-PI-BPDA (Fig. S8, previous Fig. S7)
- 2D-PI-DhTPA (Fig. S8, previous Fig. S7)
- Molecular interstitials of 2D-PI-BPDA, A-type-slip-(a-b), A-type-slip-(a), R-type and their statistical models (Fig. S11, previous Fig. S10)
- Molecular interstitials of 2D-PI-DhTPA, A-type-slip-(a-b), A-type-slip-(a), R-type and the statistical model of A-type-slip-(a) (Fig. S11, previous Fig. S10)
- Viologen-immobilized 2DP (Fig. S13, previous Fig. S12)

DOI :10.5281/zenodo.6481003

The following codes have been deposited on Zenodo:

- Python code for generating statistical models of molecular interstitials (under the same doi of the CIF files).
- Matlab script with neural network code for SAED pattern analysis and critical fluence determination (DOI: 10.5281/zenodo.6470239)

The neural network code for automated analysis of amorphous materials has been improved and expanded in the last months. The NN has been generalized to monolayer amorphous and crystalline materials, capable of handling images with strong defocus, surface contamination, illumination gradient, etc. Further development and refinement have allowed for automation in real-space polygon mapping, defect detection, boundary mapping, local strain mapping, and dopant identification. We are currently preparing a manuscript describing the details of the deep learning approach. Therefore, we have made the code available to reviewers only, which can be downloaded from <https://cloudstore.uni-ulm.de/s/763i6ZtJHNyGREd>. We sincerely hope that this could address the referee's concerns adequately.

At a couple of points the internal referencing has gone awry:

Page 6 Line 249, 250: "2D-PI" could refer to either of the two square lattice COFs that form the bulk of the study, 2D-PI-BPDA and 2D-PI-DhTPA (I assume it's the second of these). Also, the reference to Fig 3C and 3F must be incorrect, I assume they mean Fig 4C and 4F.

Similarly;

Page 6 Line 277,279,288: "Fig 4B/E/F" should be Figure 5X.

Fig S9 caption: "first-principle calculations" is a bit vague. If they used the same DFTB method as in the rest of the paper, they should say so here.

Response: We thank the referee for the great attention to detail and apologize for the mistakes in the initial submission. The abovementioned mistakes have been corrected and highlighted in the revised manuscript.

Reviewer #3 (Remarks to the Author):

This is an interesting study examining the optimization of the accelerating voltage for phase contrast HRTEM of beam sensitive 2D polymer thin films, inspired by the work of Russo and co-workers who also examined the idea of enhancing the information per electron by lowering the accelerating voltage. The authors show a significant advantage to working at 120 kV for these materials.

Response: We thank the referee for the positive feedback.

A few comments I feel would be good to address before publication:

What was not so clear to me is why exactly 120 kV turned out to be much better than 80 kV in this case. This seems likely a combination of the sample thickness and chromatic aberrations. The authors indeed seem to speculate that the drop in the measured Eel at 80 kV is due to a combination of decreased scattering mean free paths and dynamic scattering, but it would be beneficial if they elucidate the reason for the drop below 120 kV more clearly. As it is, I see no mention of chromatic aberrations at all, and one would expect this to be quite important at lower kVs. How much of a role do chromatic aberrations play?

Response: The reviewer's comments are highly appreciated. The decrease of Bragg intensity at 80 kV can be explained by the dynamic elastic scattering and inelastic scattering processes. Under the kinematic theory, the elastic scattering cross-section σ_e is proportional to $1/\beta^2$, suggesting a linear increase of ϵ_{el} with decreasing voltage⁹. The linear relation has been experimentally demonstrated on graphene¹⁰. However, the kinematical scattering is valid only for very thin specimens, for which the Bragg reflection intensity is negligibly small compared to that of the direct beam. Due to the finite thickness of the 2D polymer thin films (c.a. 15 – 60 nm, see the response to the next comment), the kinematical theory no longer applies, which is illustrated by the non-linear change of ϵ_{el} in Fig. 1e. Particularly at lower voltages where the elastic scattering mean free path is reduced, dynamic scattering leads to the intensity exchange between Bragg reflections and the direct beam¹¹. In addition, increased inelastic events at lower voltage may cause a further decrease of the elastic scattering amplitude, giving rise to reduced ϵ_{el} under 80 kV¹¹.

The discussion above has been added to the revised manuscript (Page 4, lines 153-161).

Reducing the acceleration voltage and/or increasing the sample thickness give rise to more inelastic scattering events, lowering the elastic scattering amplitude and thus phase contrast. If the inelastically scattered electrons undergo another elastic scattering process, the scattered and non-scattered parts are coherent and may provide crystal structure information¹¹. However, due to chromatic aberration, the inelastically scattered electrons are no longer focused onto the Gaussian image plane, leading to image blurring. The information transfer is limited due to the reduced temporal coherence^{12,13}. Chromatic aberration correction elevates the resolution limit imposed by the temporal coherence damping function and meanwhile enhances the image contrast due to the increased S/N ratio by refocusing the coherent inelastic electrons onto the Gaussian image plane. The enhancement of the S/N ratio in thick specimens with chromatic aberration correction has been quantitatively studied in a recent work by Russo and co-workers¹⁴.

In the revised manuscript, we have included a brief discussion on chromatic aberration correct as perspectives for future investigations (Page 4, lines 181 - 190).

The authors describe the samples as being of tens of nanometers thick, up to 60 nm. That seems quite a large range of thicknesses. The findings of Russo et al (ref 28 of the present manuscript, eg their fig 5) would seem to suggest that quite a range of voltages might be optimal for the different tens of nanometers of thickness of the "2D" sheets considered here. Do the authors disagree with this? Is 120 kV optimal for all the different thicknesses of these 2D sheets?

The referee's comment is highly appreciated. We fully agree that the optimal electron energy is thickness dependent, which has been demonstrated in the work of Russo et al¹⁰. In this work, the thicknesses have been determined via AFM and the comparison between simulated and experimental HRTEM images. AFM measurement on the edge of the films revealed a thickness of 40 – 60 nm. However, in the TEM experiment, we did find thinner regions. For instance, HRTEM images (Fig 3 and 4) were acquired from areas with a thickness ranging between 15 – 20 nm). Thickness control with better precision remains a synthetic challenge, which needs to be overcome. The optimal acceleration voltage of 120 kV in this work was experimentally determined on the 2D polymer thin films with thickness variations. Reducing the average sample thickness would undoubtedly shift the optimal energy to even lower values¹⁰. In future experimental work, we will further improve the thickness control, and explore the optimal electron energy for thinner specimens with reduced thickness variation.

In the revised manuscript, we have included the discussion of optimal energy as a function of sample thickness (Page 4, lines 177 - 180).

The comparison shown between 300 kV and 120 kV in figure 2 is interesting. Would it not be useful to include also the other voltages considered, 200 and 80 kV to help more clearly say where the advantages exist (or not) for 120 kV compared to these as well?

We thank the reviewer for the suggestion. The thickness-defocus map at 80, 120, 200, and 300 kV are included in Supplementary Fig S6. The heat maps demonstrate the image contrast enhancements of lower acceleration voltages compared to 300 kV. The results at 200 kV are similar to 300 kV. For very thin samples (under 10 nm), the contrast enhancement at 80 kV is even more prominent. The enhancement decreases to a comparable level of 120 kV as the sample thickness increases. This further demonstrates that the optimal voltage for HRTEM imaging is highly related to the sample thickness.

References

1. Nagai, A. *et al.* Pore surface engineering in covalent organic frameworks. *Nat. Commun.* **2**, (2011).
2. Li, Z., He, T., Gong, Y. & Jiang, D. Covalent organic frameworks: pore design and interface engineering. *Acc. Chem. Res.* **53**, 1672–1685 (2020).
3. Alexander, P. & Charlesby, A. Energy transfer in macromolecules exposed to ionizing radiations. *Nature* **173**, 578–579 (1954).
4. Fryer, J. R. The effect of dose rate on imaging aromatic organic crystals. *Ultramicroscopy* **23**, 321–327 (1987).
5. Mark, S. *et al.* Toward developing a predictive approach to assess electron beam instability during transmission electron microscopy of drug molecules. *Mol. Pharm.* **15**, 5114–5123 (2018).
6. Stenn, K. & Bahr, G. F. Specimen damage caused by the beam of the transmission electron microscope, a correlative reconsideration. *J. Ultrastructure Res.* **31**, 526–550 (1970).
7. Chapiro, A. Radiation Chemistry of Polymers. *Radiat. Res. Suppl.* **4**, 179 (1964).
8. Chamberlain, T. W. *et al.* Isotope substitution extends the lifetime of organic molecules in transmission electron microscopy. *Small* **11**, 622–629 (2015).
9. Egerton, R. F. Radiation damage to organic and inorganic specimens in the TEM. *Micron* **119**, 72–87 (2019).
10. Peet, M. J., Henderson, R. & Russo, C. J. The energy dependence of contrast and damage in electron cryomicroscopy of biological molecules. *Ultramicroscopy* **203**, 125–131 (2019).
11. Rose, H. Future trends in aberration-corrected electron microscopy. *Phil. Trans. R. Soc. A* **367**, 3809–3823 (2009).
12. Linck, M. *et al.* Chromatic aberration correction for atomic resolution TEM imaging from 20 to 80 kV. *Phys. Rev. Lett.* **117**, 1–5 (2016).
13. Bell, D. C., Russo, C. J. & Kolmykov, D. V. 40keV atomic resolution TEM. *Ultramicroscopy* **114**, 31–37 (2012).
14. Dickerson, J. L., Lu, P.-H., Hristov, D., Dunin-Borkowski, R. E. & Russo, C. J. Imaging biological macromolecules in thick specimens: the role of inelastic scattering in cryoEM. *Ultramicroscopy* **237**, 113510 (2022).

REVIEWERS' COMMENTS

Reviewer #1 (Remarks to the Author):

I'm pleased with the extensive reply and revisions made.

Reviewer #2 (Remarks to the Author):

I thank the authors for their response to both my and the other reviewers comments.

I appreciate (indeed, I am happy) that the authors intend to publish their Neural Network, "details in a future publication" is fine and I look forward to seeing this work.

Reviewer #3 (Remarks to the Author):

The authors have addressed my concerns in the revised manuscript and I recommend publication.